# Assessing the Impact of Road Network on Urban Landscape Ecological Risk Based on Corridor Cutting Degree Model in Fuzhou, China

**Zichun Yan, Ninglong You \*** **, Lu Wang and Chengwei Lan**

College of Landscape Architecture, Fujian Agriculture and Forestry University, Fuzhou 350002, China
\* Correspondence: nlyou@fafu.edu.cn

**Abstract:** The rapidly expanding road network has resulted in the separation of the urban ecological landscape. To assess the potential implication of the road systems on the landscape ecological risk, the corridor cutting degree model based on roadway impact zones was introduced, and the effects of the road system on the landscape pattern change were analyzed in Fuzhou City, China, in 2000, 2010, and 2020. Meanwhile, through spatial auto-correlation analysis and a geographical detector model, it was shown that there was a link between the characteristics of the road network and the temporal and spatial distribution of landscape ecological risk index, and the main determinants of landscape ecological risk were identified. The outcome indicated that (1) the intermediate cutting had the greatest impact on the ecological landscape of the four corridor cutting modes of the road network. Furthermore, the land types with a higher corridor cutting degree index were woodland, cultivated land, and grassland, accounting for 35.23%, 33.61%, and 5.95% of the total cutting areas, respectively, and the landscape fragmentation was relatively serious. (2) Fuzhou's landscape ecological risk has significantly increased over the past 20 years, with sub-high-risk and high-risk areas experiencing increases of 9.47% and 7.63%, respectively, and the spatial distribution pattern being primarily high-high and low-low clustering. (3) Corridor cutting degree index (CCI) and distance from sampling point to road (shortest distance) were two key factors that altered the geographic distribution of ecological risk in the landscape, and they showed a positive and negative connection, respectively. (4) In the geographic distribution of landscape ecological risk, the interaction between CCI and land type, or shortest distance and land type, was much higher than that of other components, with an explanation rate of more than 22%. The study findings could provide a scientific basis for integrated transportation and ecological restoration strategies in national space.

**Keywords:** roadway impact zones; Fuzhou City; landscape ecological risk index; corridor cutting degree index

## 1. Introduction

The conflict between human existence and the environment has worsened with the rapid growth of the global social economy, hence, a need to pay attention to ecological security. The intensity of human activities and natural disasters has regional and cumulative characteristics in different land-use patterns and landscape structures, which affects the balance, stability, and flow of ecosystems as well as increasing regional potential landscape ecological risk [1,2]. The creation of landscape ecological safety systems depends on an accurate assessment of landscape ecological risk, which can show the detrimental impacts of landscape patterns interacting with ecological processes when disturbed by human and natural forces [3]. The excessive exploitation of landscape resources and numerous infrastructure construction projects have emerged as primary sources of urban landscape ecological risk. Road engineering, as the world's fastest and largest infrastructure construction, is particularly visible in the division and destruction of the surrounding

natural environment, which results in ecological imbalance and raises the likelihood of harm from landscape ecological risks [4,5].

Currently, many studies evaluate landscape ecological risk using two fundamental techniques: risk index evaluation based on risk sources and sinks, and risk index evaluation based on landscape patterns [6]. Risk index evaluation based on risk sources and sinks must first identify disaster risk factors and risk sources with threats before building the "source-sink" risk structure model that is appropriate for the study area with specific risk stress factors and risk receptors [7]. Risk index evaluation based on landscape patterns no longer evaluates a certain risk factor, but comprehensively evaluates the landscape ecological risk at the regional scale by coordinating the regional landscape pattern structure and spatial distribution characteristics [8,9]. The current assessment of the ecological risk to the landscape is mostly based on the overall change in land-use and the geographical and temporal heterogeneity of that change, which is divided into two types of artificial and natural geographical units for analysis: natural units dominated by watersheds [10,11], island lakes [12,13], wetlands [8,14], and plateaus [9] as well as artificial units dominated by mining land [15], urban expansion space [16], and cultural relics [17,18]. Furthermore, some studies have built ecological risk assessments from the standpoint of land-use change and linked ecological service systems with ecological risk assessments [19,20]. However, there is little research on the main forces behind changing landscape ecological risk, especially on the quantitative evaluation of the components that influence landscape ecological risk change [21]. Meanwhile, research on the mechanism of road network on landscape ecological risk is gaining traction as a significant factor driving the transformation of urban landscape ecological risk [4,22]. Forman found that approximately 15–20% of the land space in the United States is affected by the road network [23]. In China, the affected area has reached 18.37% [24]. Road networks have an impact on the global ecological environment. In recent years, researchers have focused on road ecology. The construction of road networks has seriously interfered with the evolution of landscape patterns and migration of animal species, particularly in populated areas. Animals frequently pass away on roadways during heavy traffic [25]. The resilience of the surrounding natural environment has been diminished by the expansion of the road network, thereby increasing the ecological risk of the terrain along the path [22,26].

According to existing research, the grade, length, density, and other characteristics of the road network are important factors that cause ecological landscape separation and increase the ecological risk of regional landscapes [27,28]. However, as a typical corridor project, the cutting impact of this road network is typically disregarded in pertinent studies, despite the fact that highway invasion and the cutting of landscape areas have a considerable negative impact on the natural environment. [29]. Moreover, it leads to a shallow analysis of potential landscape ecological risk factors, which is insufficient in ensuring the liquidity and stability of the ecological environment. In light of this, the goal of this study was to explore the spatial heterogeneity of landscape ecological risk in the study region, which identifies the main causes of changes in landscape ecological risk and investigates the impact of the road network on the spatial distribution of risk [30–32]. The necessity of the study is as follows: (1) compared with the grade, length and density of the road, the corridor cutting degree index (CCI) can more directly reflect the adverse impact of the road network on the landscape pattern and ecological process, and is an excellent indicator to optimize the road characteristic variables [33]. (2) Identify the key drivers of landscape ecological risk, and comprehensively analyze the impact of the interaction between road characteristics and natural factors on landscape ecological risk, which can make up for the research on the driving force of landscape ecological risk. (3) Road characteristics have potential hazards to the landscape ecological environment. By optimizing the road characteristic variables and comprehensively analyzing the relationship between them and the landscape ecological risk, it is more helpful for researchers to develop comprehensive transportation and ecological restoration strategies for the road ecological impact mechanism.

Generally, Fuzhou has great ecological conditions, a vast diversity of species, and abundant forest landscape resources. The capital city of Fujian Province has recently constructed a number of new highways connecting the province's various regions, but this has brought about a number of ecological issues, including vegetation degradation, decreased agricultural areas, and coastal erosion along the road [34,35]. This has compelled pertinent scholars to gradually focus on landscape ecological security. Therefore, our study considered Fuzhou City as the study area and introduced a corridor cutting degree model based on roadway impact zones to analyze the effects of the road system on landscape pattern changes. Meanwhile, spatial autocorrelation analysis and a geographical detector model were used to reveal the coupling relationship between spatial and temporal dispersion of landscape ecological risk and road network characteristics, and to identify key drivers of landscape ecological risk, with the aim of providing a scientific foundation for integrated transportation and ecological restoration strategies in the national space.

## 2. Materials and Methods

### 2.1. Study Area

Fuzhou (25°15′–26°39′ N, 118°08′–120°31′ E) is on China's southeast coast, near the Taiwan Strait. As the capital of Fujian Province, it is also vital to the economic development of the southeast coast (Figure 1). With a total area of 12,153 km$^2$, the urban area is a typical estuary basin surrounded by mountains, forming a north-south mountain system dominated by Gushan, Qishan, Wuhu Mountain, and Lianhua Peak. The landform is mainly mountainous and hilly, accounting for 72.68% of the total land area of the city, with their altitude ranges from 600 to 1000 m [36]. According to the Statistical Yearbook of Fuzhou in 2022, the population of Fuzhou will reach 7.2335 million by 2021, and the highway mileage will be 11,570.42 km. The GDP in Fuzhou was 1132.45 billion yuan, representing 23.2% of the province's overall GDP. Fuzhou City has emerged as the economic pillar of Fujian Province, with a dense and developed transportation network throughout the city, and the planning of a rational road network in Fuzhou City is critical for the city's long-term development and construction. As a result, assessing the ecological risk process in the road network impact zone can provide a scientific foundation for a national spatially integrated transportation and ecological restoration strategy [37].

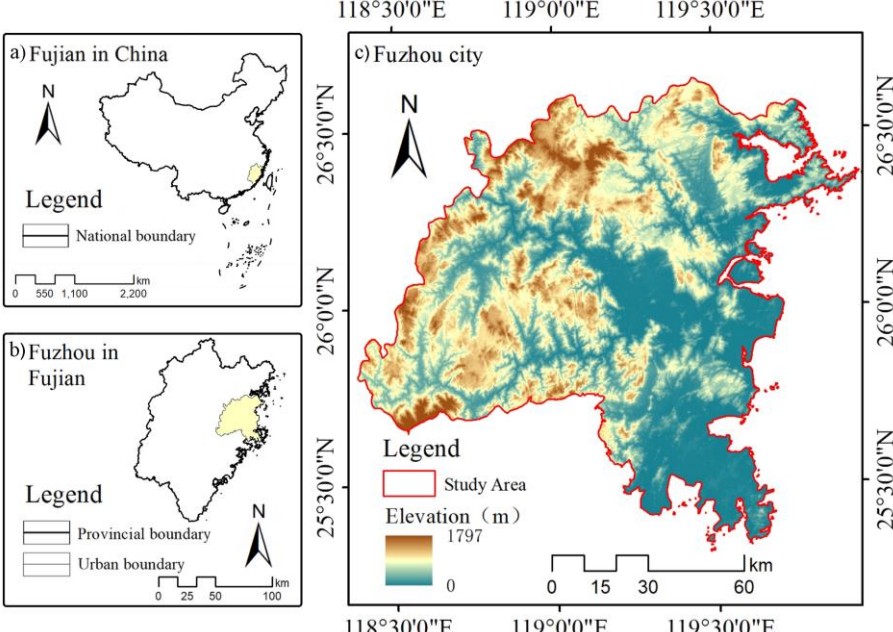

**Figure 1.** The Geographical location of Fuzhou, Fujian, China.

*2.2. Data Sources*

The land data of Globeland 30 in Fuzhou in 2000, 2010, and 2020 (30 m resolution) was chosen as the global land cover data, which came from the Chinese Ministry of Natural Resources' publication of the 30 m worldwide land cover data for 2020 (http://www.globallandcover.com/, accessed on 9 September 2022). The land types of the original data include woodland, grassland, wetland, water body, cultivated land, building land, and unused land. The road network vector data are derived from the National Basic Geographic Information Center (https://www.tianditu.gov.cn/, accessed on 20 April 2022), the People's Transport Publishing House's "China Highway Mileage Quick Search Detailed Map", and the China Map Publishing House's "China Highway Urban and Rural Road Network Atlas", and include road network vector data for three time periods: 2000, 2010, and 2020. The roadway types mostly consist of expressways, national, provincial, county, and other routes. The source of elevation information was a geospatial data cloud (http://www.gscloud.cn/, accessed on 23 August 2022). The economic data of Fuzhou comes from the statistical yearbook of Fuzhou Municipal Bureau of Statistics (http://tjj.fuzhou.gov.cn/, accessed on 16 September 2022). The traffic development data of Fuzhou comes from the Statistical Yearbook of China's Transportation (http://www.stats.gov.cn/tjsj./ndsj/, accessed on 16 September 2022).

*2.3. Methods*

Based on the calculation of "landscape ecological index", we created a landscape ecological risk index combining the landscape disturbance index, landscape vulnerability index, and landscape loss index to study the geographical relationship between the road network and landscape ecological risk, as well as the effects of driving force. Secondly, we considered the basic spatial attributes of road network, taking road density, road grade and distance from sampling point to road (referred to as "shortest distance") as basic variables, through the corridor cutting degree model to optimize the construction of road network characteristic variables. Then, using bivariate spatial auto-correlation analysis, we investigated the relationship between landscape ecological risk variables and road network characteristics and identified significant variables of road network characteristics. Finally, we introduced natural driving factors, employing geographical detector for impact assessment analysis, and suggesting coping strategies based on two levels of integrated transportation strategies and ecological restoration strategies (Figure 2).

2.3.1. The Development of Ecological Risk Assessment Framework Based on Landscape Pattern

The status of the landscape, as well as changes in the pattern of land use, can both be indicated by the landscape pattern index. Therefore, in order to research the geographic and temporally variable range and characteristics of landscape ecological risk in Fuzhou, the ecological risk assessment system based on landscape pattern index was used [38]. The calculations for the landscape pattern index were all performed using Fragestats 4.2.

(1) Landscape disturbance index

The landscape fragmentation index ($C_i$), landscape separation index ($N_i$), and landscape fractal dimension index were combined to generate the landscape disturbance index ($E_i$), which was used to reflect the degree of manmade or natural disturbance to land usage [39]. The calculating formula is as follows:

$$C_i = \frac{n_i}{A_i} \tag{1}$$

$$N_i = \frac{A}{2A_i}\sqrt{\frac{n_i}{A_i}} \tag{2}$$

$$F_i = 2\ln(p_i/4)/\ln A_i \tag{3}$$

$$E_i = aC_i + bN_i + cF_i \tag{4}$$

where $A$ is the overall area of the landscape; $A_i$ is the area of the type of landscape $i$; $n_i$ is the number of patches; $p_i$ is the perimeter of landscape type $i$; The weights of the different sorts of related landscape indices are $a$, $b$, and $c$, and $a + b + c = 1$. Combined with the actual situation of the study area, the various types of landscape indices are given the following values: $a = 0.5$, $b = 0.3$, and $c = 0.2$.

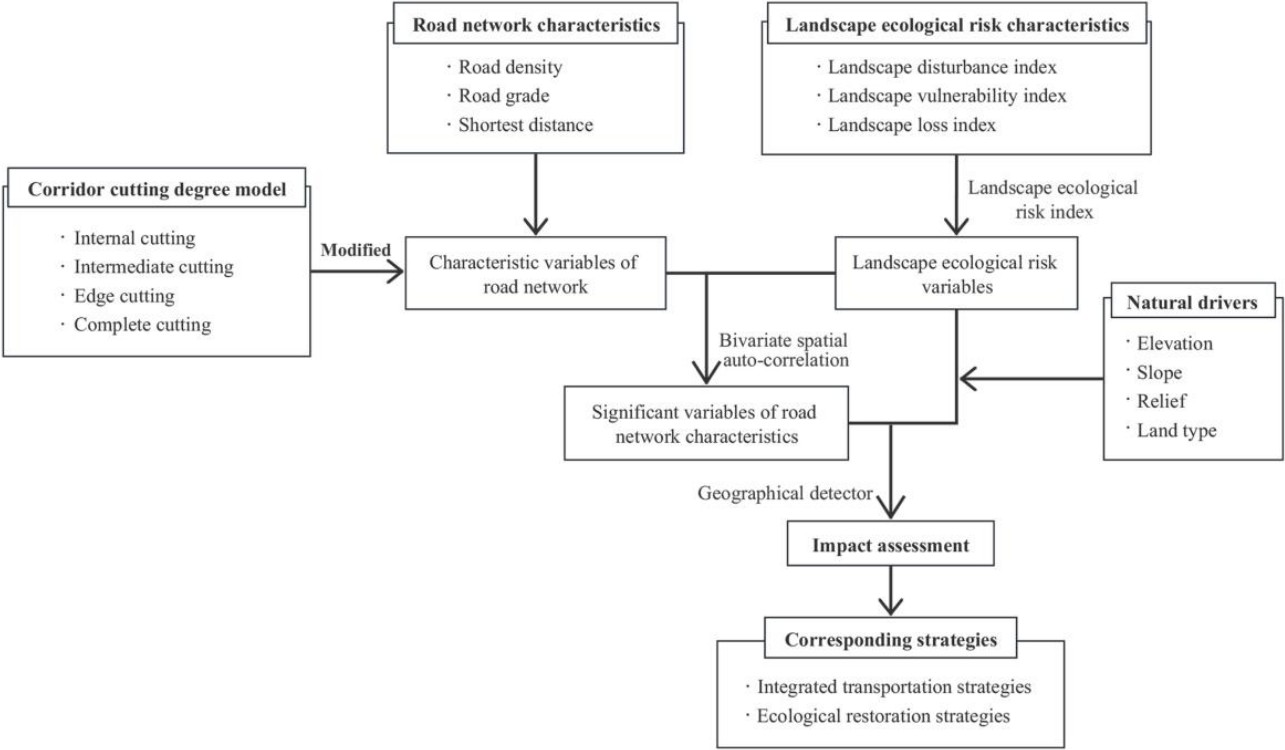

**Figure 2.** Research framework.

(2)　Landscape vulnerability index

　　The landscape vulnerability index indicates the sensitivity and fragility of a certain landscape to external disturbance. According to previous experience [4,40], the landscape vulnerability of land types with more human disturbance and weaker self-regulation ability is generally higher. Therefore, this paper adopted the expert scoring method to categorize the vulnerability index of various landscape types into seven grades, including unused land 7, wetland 6, water body 5, cultivated land 4, grassland 3, woodland 2, and building land 1. The weight of the landscape vulnerability index ($V_i$) was determined after normalization.

(3)　Landscape ecological risk index

　　In the ArcGIS 10.6 spatial analysis module, the entire study space was segmented into 2 km × 2 km evaluation units, a total of 2891, and each unit's landscape risk index underwent a separate calculation so that it became a sample for spatial interpolation analysis and was then superimposed to form a landscape risk evaluation model for the study region. The landscape ecological risk index was reclassified into the following five categories by natural breakpoint method: low ecological risk, sub-low ecological risk, medium ecological risk, sub-high ecological risk, and high ecological risk. The calculating formula is as follows:

$$R_i = E_i \times V_i \tag{5}$$

$$\text{ERI}_k = \sum_{i=1}^{N} \frac{A_{ki}}{A_k} \times R_i \tag{6}$$

where $R_i$ is the landscape loss index, which is calculated by the combination of the landscape disturbance index $E_i$ and the landscape vulnerability index $V_i$; $ERI_k$ represents the landscape ecological risk index of sample area $k$; $A_{ki}$ is the area of landscape type $I$ in the $k$th sample unit; $A_k$ is the area in the $k$th sample unit [38].

### 2.3.2. Corridor Cutting Degree Model Based on Roadway Impact Zones

Road construction interferes with the stability of ecosystems and has a potential impact on the distribution, movement, and persistence of species in the landscape, with a direct disturbance range of up to 1000 m on both sides of the road [24,41]. Therefore, according to the People's Republic of China's Highway Law, roads were classified in the form of highways, national roads, provincial roads, and county roads, and buffer zones of 1000 m, 600 m, 400 m, and 200 m were established as the extent of the roadway impact zones, respectively. The cutting pattern of the corridor can be divided into four types, which include internal cutting, intermediate cutting, edge cutting, and complete cutting (Figure 3). Roadway impact zones inside the patch are what is referred to as the internal cutting mode, without cutting it into two pieces, but the patch integrity is somewhat impacted, and the intensity is moderate. In the intermediate cutting model, the road cuts the patch into two pieces, and the energy flow and information transfer between the divided patches are completely blocked, with a more significant ecological segmentation effect. In the edge cutting mode, a small part of the edge of the patch is cut by the roadway impact zones, and the overall structure and function of the patch are not lost too much, and there is little environmental impact. The definition of complete cutting mode is the type of cutting in which the roadway impact zones completely cover the volume of one or more patches and have the greatest impact on ecological conservation [33].

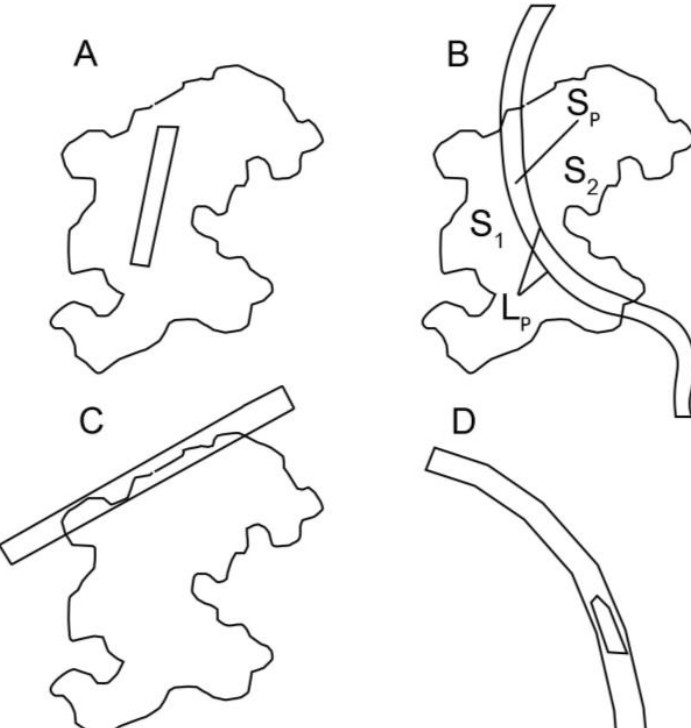

**Figure 3.** Corridor cutting degree model based on roadway impact zones. (**A**) is the internal cutting mode; (**B**) is the intermediate cutting mode; (**C**) is the edge cutting mode; (**D**) is complete cutting mode; $S_P$ is the area of roadway impact zones in patch; $S_1$ is the left area of the cutting patches; $S_2$ is the right area of the cutting patches; and $L_P$ is the adjacent side lengths between the road and the cutting surface.

Artificial corridors such as roads, bridges, and tunnels have five corridor functions in ecosystem circulation, these are connectivity, cutting, species filtering, habitat, and source of influence [42,43]. Usually, the larger the area of artificial corridors cutting patches and the adjacent side lengths, the more obvious the impact on patch connectivity. The calculation formula of the corridor cutting degree model is as follows:

$$CCI_i = \sum_{j=1}^{m} \left[ 100 \times \left( \frac{S_{ijp}}{S_{ij}} + \frac{L_{ijp}}{L_{ij}} \right) \times \frac{S_{ij}}{S} \times W \right] \tag{7}$$

where $CCI_i$ is the corridor cutting degree index of the roadway impact zones in landscape type $i$; $S_{ijp}$ is the occupation area of roadway impact zones in the $j$th cut patch of landscape type $i$; $S_{ij}$ is the area of the occupation of $j$th patch in landscape type $i$; $L_{ij}$ is the side's length of the $j$th cut patch in landscape type $i$; $L_{ijp}$ is the side's length of junction between roadway impact zones and the $j$th cut patch after the road encroaches on landscape type $i$; $S$ is the area of the evaluation units inside the research area, and in this study, a 2 km × 2 km spatial grid is used as an evaluation unit; $W$ is the weight of different cutting patterns, and edge cutting, internal cutting, intermediate cutting and complete cutting are assigned the values of 1, 3, 5 and 7, respectively [29,33]. m is the overall number of cuts in the roadway impact zones in landscape type $i$.

### 2.3.3. Spatial Correlation Analysis

An analytical technique called spatial auto-correlation analysis is used to determine whether there is a relationship between the value of an attribute on a geographical region and the value of an attribute on its nearby spatial regions [44]. There are two types of Moran's *I*: global Moran's *I* and local Moran's *I*. The former shows the average degree of aggregation of comparable attributes in the research area; the latter, which concentrates on local spatial distribution characteristics, reflects the degree of correlation between a feature of an attribute and nearby units. The values that Moran's *I* support are between −1 and +1, the correlation is considered to be positive when the value is greater than 0, and the correlation becomes more significant as the value increases [45,46]. The spatial auto-correlation analysis was run in Geoda 1.20, and Moran's *I* was calculated as follows:

$$I = \frac{\sum\limits_{i=1}^{n} \sum\limits_{j \neq i}^{n} W_{ij} \cdot (Y_i - \overline{Y}) \cdot (Y_j - \overline{Y})}{S^2 \cdot \sum\limits_{i=1}^{n} \sum\limits_{j \neq i}^{n} W_{ij}} \tag{8}$$

To detect the degree of local area correlation, local spatial autocorrelation is usually analyzed by local indicators of spatial association, (LISA). The LISA aggregation map is formed by z test, which can reflect the specific location of spatial aggregation or differentiation of research units and their neighborhood variables, and reveal the areas that have a greater impact on global correlation, its Local Moran's *I* formula was:

$$I_i = \frac{Y_i - \overline{Y}}{S_i^2} \cdot \sum_{i=1, j \neq i}^{n} W_{ij} \cdot (Y_i - \overline{Y}) \tag{9}$$

where $n$ is the number of evaluation units (2891 evaluation units were divided in this study); $Y_i$ and $Y_j$ represent the observed values of research areas $i$ and $j$, respectively; $W_{ij}$ represents the geospatial weight of the proximity relationship between units $i$ and $j$; and $S^2$ and $\overline{Y}$ represent the variance and mean values for all research areas, respectively.

To investigate the influence of road network characteristics on landscape ecological risk, four categories of (i) road density, (ii) CCI, (iii) road grade, and (iv) shortest distance were screened as independent variables and the landscape ecological risk index as dependent variables, and bivariate spatial autocorrelation was used to explore the significant

variables of road network characteristics. Bivariate spatial auto-correlation analysis can reflect the spatial aggregation relationship between two different attribute variables [45,47]. The formula for the calculation is as follows:

$$I_{b_y}^a = \frac{X_{y-\overline{x}_y}^a}{\delta_y} \times \sum_{c=1}^{n} Wac \times \frac{X_{b-\overline{x}_y}^c}{\delta_b} \tag{10}$$

where $X_y^a$ represents the value of attribute $y$ of unit $a$; $X_b^c$ represents the value of attribute $b$ of unit $c$; and $\delta_y$ and $\delta_b$ represent the variance of attributes $y$ and $b$, respectively.

### 2.3.4. Analysis of the Influencing Factors behind the Driving Force

The geo-detector model is used to determine the spatial differences between geographic elements and analyze the driving forces and affecting variables of different phenomena, as well as the interaction between multiple factors, including four detection modules with factors, risk, ecology, and interaction [32,48]. In this study, the ecological risk index of Fuzhou City landscape was used as the dependent variable, and CCI, road grade, road density, shortest distance, land type, elevation, slope, and relief were selected as drivers. Two modules—factor detection and interaction detection—were employed to pinpoint the primary variables affecting changes in ecological risk. Factor detection indicates the contribution of independent variable ($X$) to the spatial distribution of dependent variable ($Y$), and the specific formula is as follows:

$$q = 1 - \frac{1}{n\sigma^2} \sum_{h=1}^{L} N_h \sigma_h^2 \tag{11}$$

where $q$ represents the contribution importance of the driver factor to the change in the landscape risk index, with larger values representing a higher contribution; $n$ represents the quantity of evaluation units in the overall research region; $\sigma$ and $\sigma_h$ represent the overall region variance and category $h$ variance, respectively; $L$ is the category of the quantity of variables; $h = 1, 2, \ldots$ for a specific type; $N_h$ is the number of graded evaluation units of each type of data.

The detection of interaction can identify the interaction between the two risk factors to determine if the interaction of the two risk variables would strengthen or diminish the explaining ability of dependent variable $Y$ [49]. The interaction between the two factors is as follows (Table 1):

**Table 1.** Two factor interactive detection relationship [49].

| Basis of Judgement | Interaction Relation |
|---|---|
| $q(X1 \cap X2) < Min(q(X1), q(X2))$ | Nonlinear weakening |
| $(q(X1), q(X2))_{min} < q(X1 \cap X2) < (q(X1), q(X2))_{Max}$ | Single-factor nonlinear weakening |
| $q(X1 \cap X2) > Max(q(X1), q(X2))$ | Bifactor enhancement |
| $q(X1 \cap X2) = q(X1) + q(X2)$ | Two-factor independence |
| $q(X1 \cap X2) > q(X1) + q(X2)$ | Nonlinear enhancement |

$q(X1 \cap X2)$ is their interaction, $Min(q(X1), q(X2))$ is their minimum value, $Max(q(X1), q(X2))$ is their maximum value, and $q(X1) + q(X2)$ is their sum.

## 3. Results

### 3.1. Corridor Cutting Effect Based on Roadway Impact Zones

In the past 20 years, Fuzhou's transportation network has grown rapidly from 3177.88 km to 4642.81 km, and the expansion mileage of expressway alone has reached 1134.66 km. The corridor cutting effect of roads on the landscape is becoming more and more significant over time and with the expansion of the road network (Figure 4). The CCI grade was then divided into the following five classes: low, sub-low, medium, sub-high, and high. In 2000, the regions with the sub-high grade of CCI were mainly concentrated in the northwest and

the north–south axis, the urban road traffic is still in its infancy, and the landscape is not significantly affected by the corridor cutting effect. In 2010, many expressways in Fuzhou were opened, and the CCI grade in the central area of the city began to rise gradually and began to expand from the east to the west. In 2020, the CCI grade in the southern region of Fuzhou will be significantly improved, the focus of urban road construction will shift from the east to the south, and the corridor cutting effect will also be significantly improved.

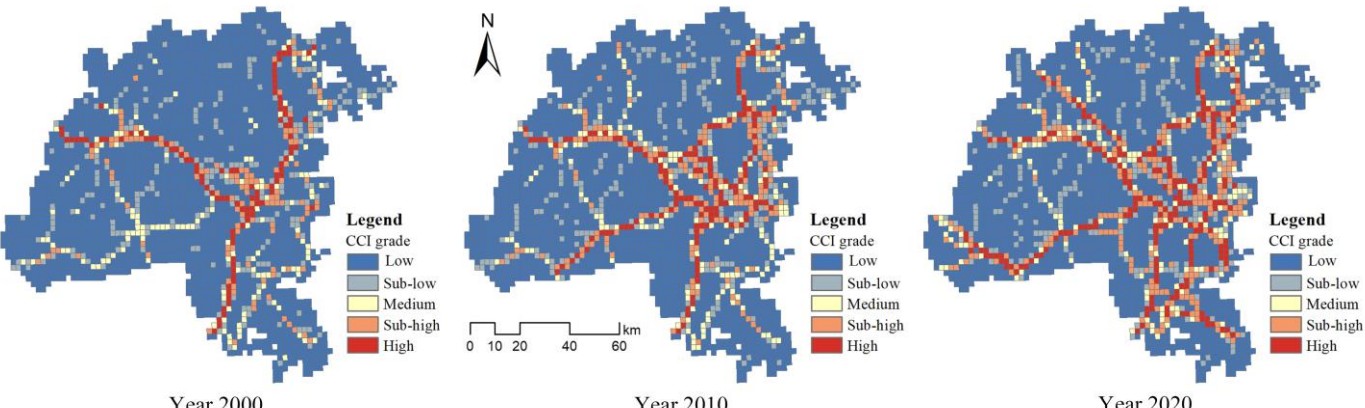

**Figure 4.** Distribution of CCI of Fuzhou from 2000 to 2020. The corridor cutting degree index is reclassified into the following five categories by natural breakpoint method: low ecological risk, sub-low ecological risk, medium eco-logical risk, sub-high ecological risk, and high ecological risk.

The number of cutting times rose over time in the various types of landscape, among which the number of edge cutting time was the largest. In 2020, the total number of edge cutting reached 2919 (Figure 5), and the three types of land with the highest number of edge cutting time were woodland, cultivated land, and grassland, with the number of cuts being 906, 673, and 680, respectively. The number of intermediate cutting types increased significantly, especially from 2000 to 2010, the total number of cuttings increased from 1537 to 2493, with an increase of 62.20%. The number of intermediate cuttings of woodland in each landscape type changed significantly, from 231 to 492 in 20 years, with an increase of 112.99%. The number of complete cutting types was the least, and the change is not obvious.

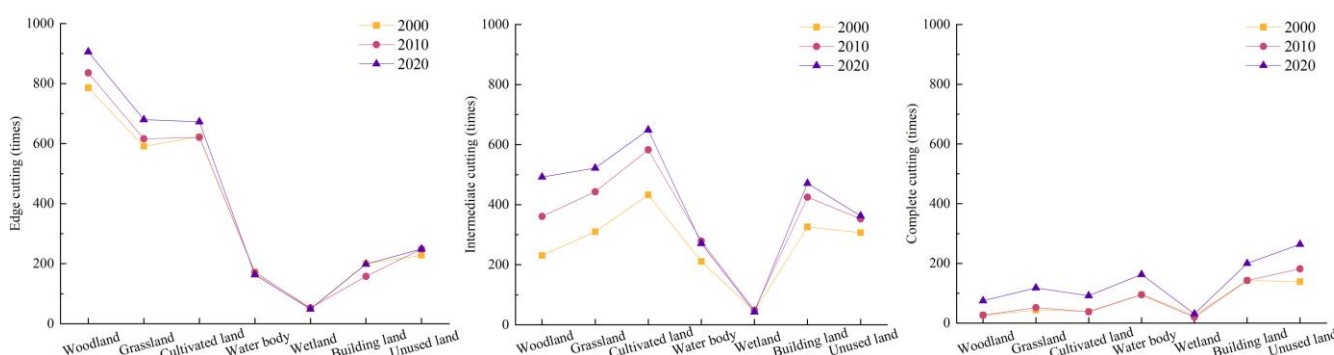

**Figure 5.** The number of times on different land types in the study area are cut by roadway impact zones.

Combined with the road cutting areas and adjacent side lengths (Table 2), the cutting area and adjacent side length of different cutting patterns were obviously different. The total cutting area increased from 916.08 km² to 1860.10 km² in 20 years, with an increase of 103.05%, while the adjacent side lengths increased from 15,378.34 km to 27,594.78 km, with an increase of 79.43%. This shows that the focus of road construction has shifted from urban centers to districts and counties, and more ecologically sensitive areas have begun to

be eroded by roads. The three landscape types, woodland, cultivated land, and grassland, were significantly affected by intermediate cutting, and the cutting areas were 625.23 km$^2$, 655.45 km$^2$ and 110.65 km$^2$, respectively, by 2020, and the adjacent side lengths reached 8344.62 km, 7824.78 km and 5701.62 km, respectively. The edge cutting area of grassland only accounted for about 6% of the total cutting areas, but the impact of road cutting was far greater than that of other land types (Appendix A: Table A1). The cutting areas and adjacent side lengths of the edge cutting type and the complete cutting type increased relatively little. The CCI values of different landscape types in descending order were cultivated land, grassland, woodland, building land, water body, wetland, and unused land.

**Table 2.** Cutting area and length of adjoining sides under different road cutting modes.

| Cutting Mode | | Time (Year) | Land Use Type | | | | | | |
|---|---|---|---|---|---|---|---|---|---|
| | | | Woodland | Grassland | Cultivated Land | Water Body | Wetland | Building Land | Unused Land |
| Cutting area (km$^2$) | Edge cutting | 2000 | 245.40 | 33.99 | 195.88 | 16.83 | 4.00 | 31.38 | 2.09 |
| | | 2010 | 288.93 | 35.90 | 178.15 | 16.99 | 5.09 | 27.87 | 1.76 |
| | | 2020 | 329.84 | 37.87 | 156.35 | 19.72 | 4.72 | 43.53 | 1.41 |
| | Intermediate cutting | 2000 | 197.57 | 55.48 | 450.33 | 51.81 | 13.21 | 135.63 | 12.05 |
| | | 2010 | 382.22 | 90.32 | 597.87 | 63.94 | 14.54 | 215.73 | 13.02 |
| | | 2020 | 625.23 | 110.65 | 655.45 | 102.58 | 12.00 | 343.22 | 10.97 |
| | Complete cutting | 2000 | 7.18 | 2.65 | 21.45 | 8.73 | 1.71 | 23.18 | 2.72 |
| | | 2010 | 10.50 | 3.12 | 24.11 | 7.95 | 1.32 | 25.63 | 2.85 |
| | | 2020 | 30.50 | 12.55 | 51.83 | 26.03 | 2.45 | 64.31 | 4.71 |
| The adjacent side length(km) | Edge cutting | 2000 | 5502.42 | 2028.96 | 3140.70 | 383.16 | 100.26 | 530.52 | 185.16 |
| | | 2010 | 6150.72 | 2159.46 | 3124.80 | 387.72 | 116.88 | 439.98 | 161.82 |
| | | 2020 | 6383.04 | 2270.70 | 3030.90 | 359.76 | 104.76 | 599.82 | 138.90 |
| | Intermediate cutting | 2000 | 3387.12 | 3054.48 | 5058.36 | 1018.20 | 230.76 | 1805.28 | 824.64 |
| | | 2010 | 5849.40 | 4785.24 | 6756.06 | 1354.38 | 240.54 | 2642.64 | 947.58 |
| | | 2020 | 8344.62 | 5701.62 | 7824.78 | 1367.10 | 188.64 | 3346.68 | 821.34 |
| | Complete cutting | 2000 | 134.70 | 163.02 | 274.02 | 271.38 | 51.00 | 442.44 | 213.78 |
| | | 2010 | 197.10 | 198.00 | 316.68 | 240.90 | 39.00 | 422.04 | 241.50 |
| | | 2020 | 605.04 | 699.54 | 689.70 | 506.22 | 69.06 | 744.60 | 361.26 |
| CCI | | 2000 | 30,893.48 | 116,802.03 | 107,266.61 | 13,946.28 | 3922.013 | 15,857.29 | 3491.29 |
| | | 2010 | 50,569.35 | 155,852.72 | 160,449.51 | 17,765.98 | 5450.517 | 23,461.65 | 3751.99 |
| | | 2020 | 100,620.51 | 195,586.41 | 225,109.60 | 21,020.34 | 6704.392 | 28,550.62 | 889.46 |

Note: cutting area is the cutting surface between roadway impact zones and different land types; the adjacent side length is the lengths between roadway impact zones and the cutting surface; CCI is the corridor cutting degree index of roadway impact zones.

### 3.2. Impact of Road Network on Ecological Risk in the Landscape

3.2.1. Spatial and Temporal Variation of Ecological Risk in the Landscape

The landscape ecological risk index was divided into the following five classes: low, sub-low, medium, sub-high, and high (Figure 6). The average value of the ecological risk of the Fuzhou landscape increased from 0.045 to 0.051 and subsequently decreased to 0.049 between 2000 and 2020, showing a rising tendency. According to our analysis, both the sub-high and high ecological risk areas grew at rates of 9.47% and 7.63%, respectively. From 2000 to 2010, the sub-high ecological risk area increased by a larger amount, from 21.58% to 29.92%, with an area increase of 954.73 km$^2$. Moreover, the area of high ecological risk

increased by 666.37 km², raising its share by 5.83%. From 2010 to 2020, the area of medium ecological risk decreased considerably from 37.39% to 29.14%, and the area decreased by 943.83 km² (Table 3). However, the sub-high and high-risk areas increased, but the increase was relatively small, only increasing by 1.13% and 1.80%, respectively. The geographical heterogeneity of the local zone was evident from the perspective of the spatial change in the landscape risk index, and the ecological risk zone above the sub-high level gradually extended from the city's core to the surrounding zone. The ecological risk of Changle District, the eastern sub-city center of Fuzhou, has changed greatly, and the ecological risk in the central-eastern part of Changle District is changing from medium- to high-risk. As the pivot of Fuzhou's eastward development, the district has an efficient and sound transportation system, connecting Luoyuan Port to the north and Pingtan Experimental Base to the south, becoming the second largest economic zone in Fuzhou in addition to the central city. However, its ecological environment is under great pressure due to rapid economic development. In addition, the dense traffic network of Lianjiang County and Luoyuan County had a notable increase in the ecological risk to the landscape. Overall, the ecological risk index tends to be higher in urban zones with dense traffic, where the areas of low-risk and medium-risk zones gradually decrease, and the area of high-risk zones gradually increases and shrinks toward the urban center.

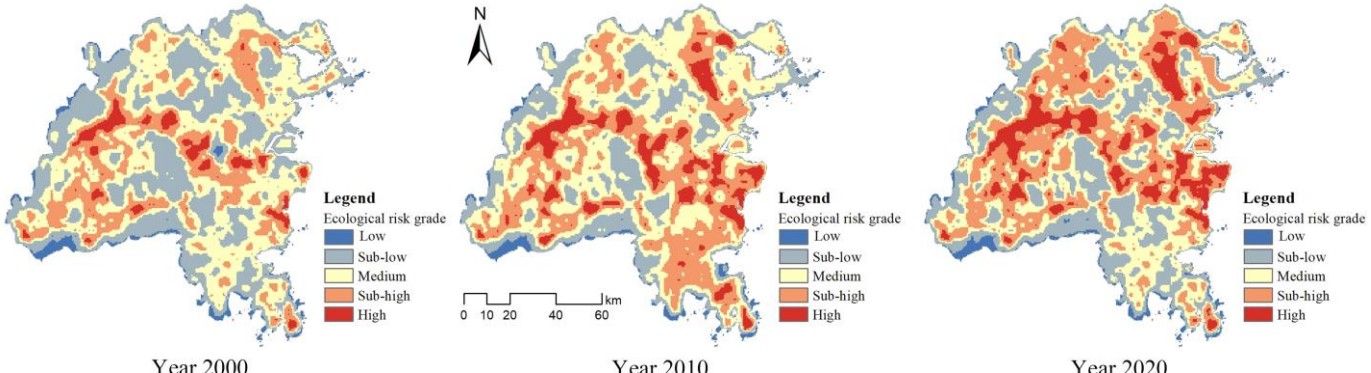

**Figure 6.** Distribution of landscape ecological risk index in Fuzhou from 2000 to 2020. The landscape ecological risk index is reclassified into the following five categories by natural breakpoint method: low ecological risk, sub-low ecological risk, medium ecological risk, sub-high ecological risk, and high ecological risk.

**Table 3.** Landscape ecological risk area and proportion in 2000, 2010 and 2020.

| Ecological Risk Grade | Year 2000 | | Year 2010 | | Year 2020 | |
|---|---|---|---|---|---|---|
| | Area (km²) | Proportion (%) | Area (km²) | Proportion (%) | Area (km²) | Proportion (%) |
| Low ecological risk | 332.66 | 2.91 | 280.97 | 2.46 | 318.24 | 2.78 |
| Sub-low ecological risk | 3860.75 | 33.75 | 2344.09 | 20.49 | 2915.52 | 25.49 |
| Medium ecological risk | 4329.85 | 37.85 | 4277.11 | 37.39 | 3333.28 | 29.14 |
| Sub-high ecological risk | 2468.22 | 21.58 | 3422.95 | 29.92 | 3552.01 | 31.05 |
| High ecological risk | 446.95 | 3.91 | 1113.32 | 9.73 | 1319.39 | 11.53 |

### 3.2.2. Spatial Correlation of Ecological Risk in the Landscape

The ecological risk data from the three time periods were subjected to a spatial auto-correlation analysis, which produced global Moran's *I* values for 2000, 2010, and 2020 of 0.53, 0.56, and 0.57, respectively, which passed the significance test with *p* values less than 0.01 and *Z* values greater than 2.58 (Table 4). The geographical distribution of landscape ecological risk in Fuzhou exhibited a clear positive association, as shown by the fact that all three global Moran's *I* indices were positive. Areas with high ecological risk values in the

study region also had high values in their neighboring areas, and vice versa. The spatial autocorrelation of landscape ecological risk throughout Fuzhou City was increasing, and Moran's *I* clearly showed an increasing tendency over time.

**Table 4.** Spatial autocorrelation analysis of landscape ecological risk in fuzhou from 2000 to 2020.

| Time (Year) | Moran's *I* | *p* | *Z* |
|:---:|:---:|:---:|:---:|
| 2000 | 0.53 | 0.001 | 51.23 |
| 2010 | 0.56 | 0.001 | 52.54 |
| 2020 | 0.57 | 0.001 | 52.50 |

Further analysis of the local spatial correlation of landscape ecological risk in Fuzhou City shows that high-high and low-low aggregation forms predominated in Fuzhou City's spatial distribution of landscape ecological risk, and the distribution characteristics remained basically the same in different periods (Figure 7). In the city core of Fuzhou, the high-high aggregation is the primary form, the new coastal city of Changle District, the central part of Minhou, the central-eastern part of Minqing County, the northern part of Lianjiang County, the southern part of Luoyuan County, and the central-northern part of Yongtai County, all of which have developed transportation road networks passing through. The ecological risk level of nearby places is also higher (Figure 8). The high-risk areas with high aggregation increased from 479 to 532 in 20 years, and the positive spatial correlation increased significantly. The low-low aggregation form is primarily in the border areas of districts and counties, the north and south of Minhou County, and the south of Lianjiang County, which are more remote, have a single landscape type, and are far from urban centers. The risk area of low-low agglomeration increased from 360 to 394 blocks with time. The number of agglomerations in the central part decreased significantly, and the agglomeration phenomenon in the edge zone of the city area increased significantly.

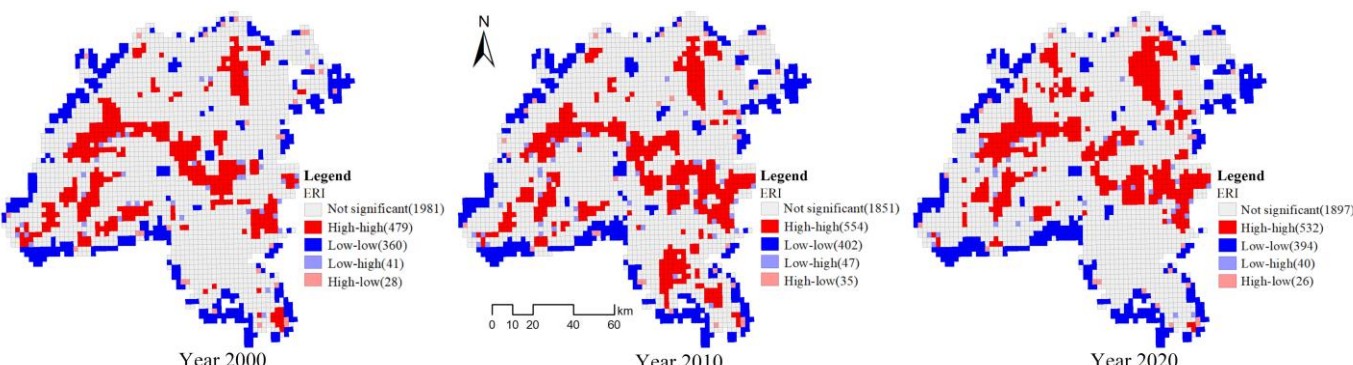

**Figure 7.** Local spatial auto-correlation aggregation map (LISA) of landscape ecological risk in Fuzhou.

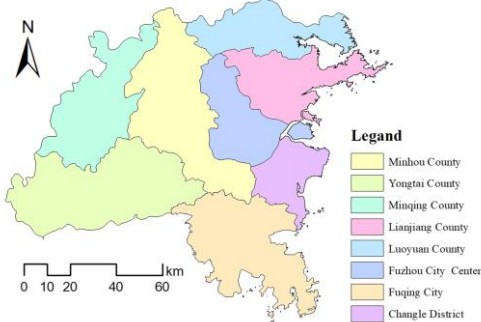

**Figure 8.** All districts and counties in Fuzhou.

### 3.2.3. Significant Road Network Characteristics Analysis

The four categories of (i) road density, (ii) CCI, (iii) road grade, and (iv) shortest distance and landscape ecological risk bivariate global Moran's *I* values passed $p < 0.01$ significance test, and the absolute value of the Z-value was significantly higher than 2.58 (Table 5). The bivariate global Moran's *I* values of the four categories of indicators and landscape ecological risk all increased and then decreased from 2000 to 2020 because the rapid expansion of the road network in Fuzhou led to the multiplication of the pressure on the surrounding ecological environment, thus increasing the burden of landscape ecological risk. In recent years, Fuzhou has gradually attached importance to urban ecological environment restoration and advocated vigorous afforestation and no blind expansion of road projects, thus reducing the degree of landscape ecological risk. Road density, CCI, road grade, and landscape ecological risk showed an obvious positive correlation in the spatial layout. Along with the construction of the road, its landscape ecological risk increased, while the shortest distance and landscape ecological risk spatial distribution showed an obvious negative correlation. This is because the county is rural far from the urban center, its road network density is sparse, and the surrounding ecological environment has not been affected by human activities. Therefore, the ecological risk decreased with increasing distance from the road. In terms of correlation performance, the shortest distance and CCI were the two largest significant variables of road characteristics affecting the spatial variables of landscape ecological risk, and the bivariate Moran's *I* of landscape ecological risk with the shortest distance could reach up to −0.37 at most, while the bivariate Moran's *I* with CCI could reach up to 0.24 at most. In summary, there is some order for the spatial distribution of landscape ecological risk and there is an obvious correlation with the construction of road projects. The high aggregation of road networks will contribute to the intensification of the cutting impact of corridors, which will eventually lead to a relative increase in landscape ecological risk in its surrounding areas.

**Table 5.** Results of bivariate spatial correlation analysis between four categories of road factors and landscape ecological risk.

| Independent Variable | Time (Year) | Landscape Ecological Risk Index | | |
| --- | --- | --- | --- | --- |
| | | Moran's *I* | *p* | *Z* |
| Road density (i) | 2000 | 0.17 | 0.001 | 20.94 |
| | 2010 | 0.22 | 0.001 | 27.49 |
| | 2020 | 0.19 | 0.001 | 23.71 |
| CCI (ii) | 2000 | 0.16 | 0.001 | 20.01 |
| | 2010 | 0.24 | 0.001 | 28.09 |
| | 2020 | 0.21 | 0.001 | 24.16 |
| Road grade (iii) | 2000 | 0.17 | 0.001 | 20.81 |
| | 2010 | 0.18 | 0.001 | 22.44 |
| | 2020 | 0.17 | 0.001 | 21.18 |
| Shortest distance (iv) | 2000 | −0.32 | 0.001 | −36.99 |
| | 2010 | −0.35 | 0.001 | −40.73 |
| | 2020 | −0.37 | 0.001 | −41.97 |

Note: Moran's *I* is bivariate global Moran's *I*; *p* is a significance test method in statistics, and it is generally considered that when the $p < 0.01$ is considered to be statistically significant; *Z* represents a multiple of the standard deviation, and it is generally considered statistically significant that the absolute value of *Z* is greater than 2.58.

### 3.3. Identification of Key Drivers

The four human factors of road density, road class, CCI, and shortest distance in the above analysis were combined with the four natural factors of land type, elevation, slope,

and relief to synthesize the main drivers of landscape ecological risk in Fuzhou. The three factors that had the greatest influence on landscape ecological risk were elevation, CCI, and the shortest distance (Table 6). From 2000 to 2010, elevation had a significant effect on landscape ecological risk, with its Q-value increasing from 0.14 to 0.23 and contributing the most to landscape ecological risk; CCI and shortest distance the Q-values have also significantly increased to 0.20 and 0.19. From 2010 to 2020, the contribution of each factor to the landscape risk index declined, and the Q-value of the elevation dropped to 0.12, while the Q-value of CCI and shortest distance exceeded the elevation by 0.15 and 0.16, respectively. Other road-related indicators also exceeded the Q-value contribution of natural factors.

**Table 6.** Contribution degree of landscape risk factors.

| Time (Year) | Analysis Results | Driving Factors | | | | | | | |
|---|---|---|---|---|---|---|---|---|---|
| | | Land Type | Elevation | Slope | Relief | Road Density | Road Grade | CCI | Shortest Distance |
| 2000 | Q value | 0.09 | 0.14 | 0.07 | 0.08 | 0.09 | 0.10 | 0.11 | 0.12 |
| 2010 | Q value | 0.18 | 0.23 | 0.13 | 0.14 | 0.17 | 0.18 | 0.20 | 0.19 |
| 2020 | Q value | 0.11 | 0.12 | 0.07 | 0.08 | 0.12 | 0.12 | 0.15 | 0.16 |

Note: land type, elevation, slope and relief are natural factors; road density, road grade, CCI and Shortest distance are human factors.

The findings of the interaction detection indicate that the two-factor interaction greatly outweighed the influence of a single factor (Figure 9), and the interaction relationship between the two factors mainly shows "bifactor enhancement" and "nonlinear enhancement," indicating that the ecological risk of the Fuzhou landscape is coordinated by both human and natural factors. In 2000, the factors of elevation and land type had the highest interactive influence. The two interactions were "bifactor enhancement," while the interactive influence of land type and road density, land type, and CCI were all "nonlinear enhancement," and their potential influences of bivariate interaction was greater than the interactive influence of elevation and land type. The explanatory power of land type as a single factor is weak, but it is greatly enhanced when combined with the road factor. In 2010, the interactive influence of the factors significantly increased, the highest explanatory power of the interactive influence of shortest distance and elevation reached 0.32, and the interaction relationship showed "bifactor enhancement." In 2020, the interactive influence of elevation and road gradually weakened, and the explanatory power of the interactive influence of land type and shortest distance and CCI reached 0.23 and 0.22, respectively. These have become the main drivers of landscape ecological risk. Combined with the two-factor interaction, the interaction of elevation and land type has a high explanatory power for landscape ecological risk in the early stage, but the explanatory power of the road factor gradually exceeds that of the natural factor as time changes. By 2020, the explanatory power of the interactive influence of land type and CCI, and the interactive influence of land type and shortest distance were significantly improved, and the explanation rate reached more than 22%.

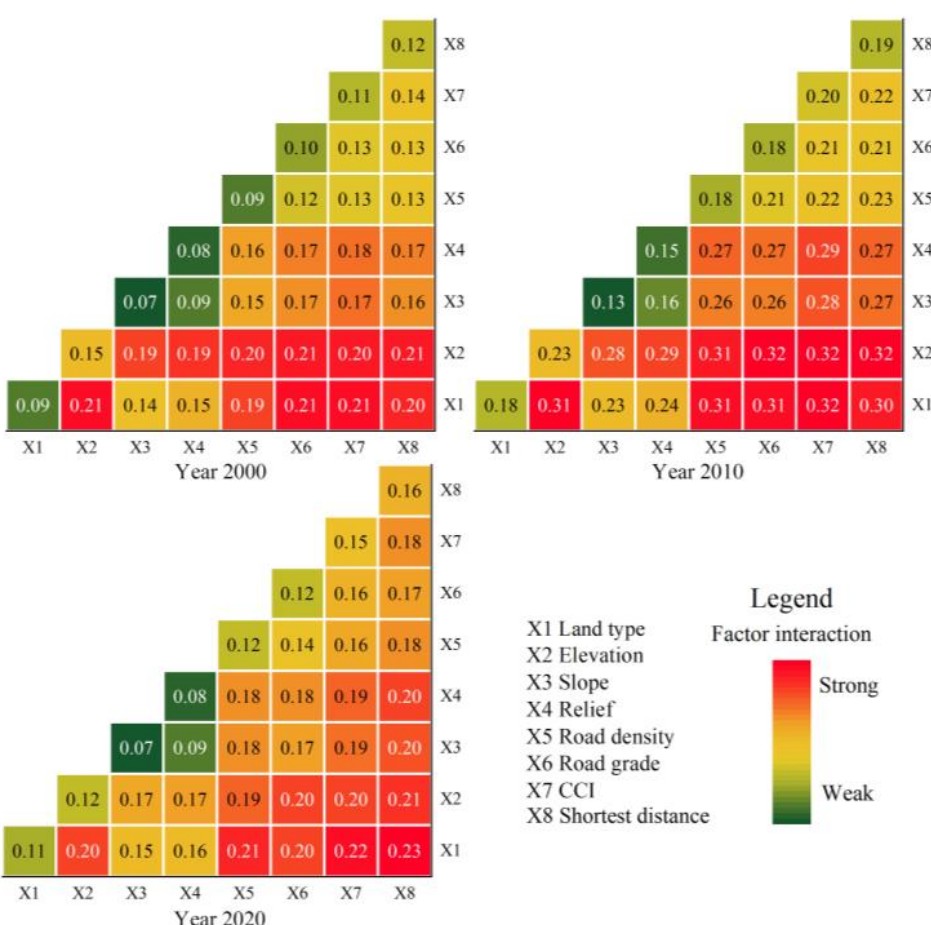

**Figure 9.** Analysis of interactive detection strength of driving factors from 2000 to 2020.

## 4. Discussion

### 4.1. Impact of Road Cutting Effect on Ecological Risk in Urban Landscape

This study referred to the four primary types of road cuttings to analyze the spatial impact of road cutting on the landscape. It then calculated the corridor cut degree index based on the roadway impact zones with a 2 km × 2 km evaluation unit, which effectively reflected the impact of road cutting on the change in land-use types [29]. Previous studies on the ecological impact of roads have generally attributed the basic characteristics of roads as the main factors causing landscape imbalance, such as road density, road length, and road grade, and have ignored the corridor cutting effect of roads [4,50]. The road network formed by the interweaving and convergence of linear artificial corridors has a significant segmentation and isolation effect on the landscape space, and the effect of this cutting influence varies in different land-use types [51].

In the roadway impact zone analysis, woodland, cultivated land, and building land occupied a larger area, whereas grassland only had a spatial share of approximately 6% in the roadway impact zones. However, the CCI of building land and woodland was lower than that of grassland, which indicates that land types with large areas in the roadway impact zones were not necessarily more affected by road cutting. A grassland area is not as ecologically restorative as a woodland area and is easily eroded by roads, because grasslands are more scattered around the city. Unlike woodland and cultivated land concentrations, this makes it difficult for grasslands to self-repair after being disturbed by external factors [52,53]. The intermediate cutting mode of roadway impact zones in Fuzhou was the most obvious, and the number of cutting times, cutting areas, and adjacent side lengths of the intermediate cutting type increased substantially between 2000 and 2020. Large urban areas were occupied by road networks, and the intermediate cutting effect

gradually became dominant. As the road network expands and covers a larger area, the risk of road damage in the ecological environment gradually increases.

This research shows that the landscape ecological risk in Fuzhou is not distributed haphazardly, and there is a clear spatial convergence relationship in part of the space, with the high-risk areas concentrated in the central part of the city and the risk index significantly reduced along the municipal border zone, which highly overlaps with the developed and dense areas of the road network. Bivariate spatial autocorrelation analysis found that landscape ecological risk had a significant spatial correlation with human factors, showing a positive correlation with CCI, road grade, and road density, and a negative correlation with the shortest distance to the road, which was consistent with Lu et al.'s research on the impact of roads on landscape patterns [54]. Elevation, CCI, and shortest distance had the greatest influence on the spatial and temporal distribution of ecological risk in the Fuzhou landscape, and the influence of human factors have gradually increased and surpassed the driving influence of natural factors over the past 20 years. However, with the development and expansion of cities and the improvement of infrastructure technology, relatively sparsely populated remote areas have gradually developed, and the impact of roads on the surrounding environment has gradually started to expand. Under interactive detection analysis, the impact of natural factors, such as elevation, on landscape ecological risk is no longer significant, while CCI and the shortest distance have become the key human factors that affect landscape ecological risk and have a significant interactive relationship with land use.

### 4.2. Integrated Transportation and Ecological Restoration Planning Strategy

In recent years, the focus of Fuzhou's future urban development has shifted, relying on the new coastal town in Changle District to develop modern service industries to shape the ecological, humanistic, and landscape coastal urban development belt. It vigorously constructs the land and sea development and protection pattern of "One Belt, Four Bays" and focuses on regional coordination and land and sea coordination [34]. To speed up the connection between the main urban area and the suburban area, Fuzhou City has built many new highways to connect the east and west ends, which gradually affect the surrounding landscape space such as forests, grasses, and cultivated land while promoting economic development. This has resulted in the intensification of ecological risks, while the remote areas at the edge of the city, with inconvenient transportation environments, less population circulation, and more undulating terrain, are less subject to human interference factors, hence their ecological environment conditions are superior to those around the city [55]. Therefore, this study proposes the following recommendations from the perspective of integrated transportation and ecological restoration planning:

(1)　Road Engineering Construction and Operation

Urban road design and planning should be timely to avoid all kinds of environmental protection targets and environmentally sensitive points or make full use of topographic and geomorphological conditions that do not destroy the original ecological environment and reduce earth excavation, blasting, and other construction projects that induce geological disasters. In Fuzhou, cultivated land, woodland, and grassland were significantly affected by road cutting. Sharp turns, steep slopes, and climbing lanes were avoided on road sections near these areas. High, flat, and vertical roads should be used to reduce direct contact between road construction and the ecological environment. This can also prevent species from dying across roads and ensure the connectivity of biological channels. To avoid excessive encroachment on land space, which makes it difficult to maintain ecological self-restoration, it is necessary to tightly regulate the site selection for road engineering construction to avoid roads directly crossing ecologically sensitive areas and to ensure the connectivity of ecological spaces [56,57].

(2)　Natural resource protection and resilient development

In the future construction of Fuzhou City, it is necessary to focus on protecting the ecological function areas of land and sea to ensure sustainable economic construction, and to plan multiple green wedges in the central city, coastal new town, and Sanjiang port to prevent unlimited urban expansion. The distribution of landscape ecological risks in Fuzhou is closely related to the urban development. With the expansion of the road network, ecological risks around the city have increased significantly. Therefore, the protection of natural resources and urban ecology has become the key to mitigating increased local risks. First, the areas along the roads are strictly controlled to avoid crossing the original ecological environment, such as woodlands and grasslands, destroying the overall function of the ecosystem, improving the forest structure, and strengthening the overall function of the forest ecosystem. Second, strictly adhering to the 'red line' of cultivated land protection and focusing on the protection of Fuqing City, Changle District, Lianjiang County, Luoyuan County, and other concentrated areas of cultivated land, the efficient development of abandoned land reclamation, moderate development of cultivated land, and reserve land resources. Buffer zones should be established according to different road grades to avoid nature reserves and ecologically sensitive areas that are too close to the road, thus leading to fragmentation of landscape space [58,59].

(3) Slope ecosystem restoration

Serious soil erosion along road zones leads to serious degradation of the surrounding ecosystem and a reduction in species richness. Fuzhou has rich vegetation resources, which can effectively mitigate the impact of road cutting on land types in Fuzhou through vertical greening and slope greening. To alleviate the impact of road cutting, applicable grass species or seedlings should be selected according to different slope types, combining trees, shrubs, grasses, flowers, and vines to form an all-round three-dimensional greening and protection effect combining shallow and deep protection [60]. The selected vegetation should be mainly local native plants with near-natural growth, which can be highly integrated with the surrounding landscape and promote the improvement of ecological self-healing ability. In addition, a protective forest belt should be set up around the slope to reduce noise interference caused by traffic on the side slope. Paying attention to ecological protection along the road can improve the diversity of the ecosystem around the road, strengthen its ecological resistance, and reduce the impact of road damage.

### 4.3. Limitations and Prospects of the Study

Quantifying the impact of road cutting is realized by the corridor cutting degree model based on roadway impact zones, which is the ecological effect zones formed between the road space and the surrounding ecological environment, and the width of roadway impact zones should be different in different geographical environments due to the influence of geographical conditions. The roadway impact zones in this study are divided according to international standards by the construction of road grade and do not construct suitable roadway impact zones based on the geographic features of the research area, so there are certain limitations. In the future, it will be considered to modify the road influence area in combination with the resistance coefficient of the study area. In addition, the landscape ecological risk evaluation is constructed based on the landscape pattern index, which focuses on the static landscape pattern analysis and pays insufficient attention to the dynamic changes of ecosystem and landscape structure function, and the index construction relies on expert experience and has strong subjectivity. Principal component analysis is a statistical analysis method that converts multiple variables into a few principal components through dimension reduction technology. It can objectively reflect most of the information of the original variables and can be considered to build a landscape ecological risk index. The assessment of landscape ecological risk is to prevent possible hidden dangers and should take into account its impact on human life, production, and ecology, ultimately providing strategies to serve human well-being. Research limited to the ecological level cannot provide effective practical help. Therefore, in the future, we can try to introduce the

value of ecosystem services and further explore the impact of road networks on ecosystems by integrating social, tourism, and economic factors.

## 5. Conclusions

This study uses a corridor cutting degree model of roadway impact zones to evaluate the possible effects of road networks on landscape ecological risk. Through spatial auto-correlation analysis and a geographical detector model, we examined the relationship between the spatial and temporal distribution of landscape ecological risk and the characteristics of the road network and identified the major factors influencing this risk in Fuzhou. The findings of this study can serve as a sound scientific foundation for national space planning that integrates transportation and ecological restoration. The following are the key findings of the study:

(1) The corridor cutting effect of roads on landscape types increases with increased road network area, and the intermediate cutting effect of the road network is the most significant. Woodland, cultivated land and grassland are the land types with high corridor cutting degree index;

(2) In the past 20 years, the area of the sub-high and high ecological risk areas in Fuzhou continued to increase, increasing by 9.47% and 7.63%, respectively. The ecological risk in the traffic intensive areas was generally high, and the spatial distribution pattern was mainly high-high and low-low;

(3) The bivariate Moran's *I* of landscape ecological risk and shortest distance can reach up to $-0.37$ at most, and the bivariate Moran's *I* of landscape ecological risk and CCI can reach up to 0.24 at most. The shortest distance and CCI are the two factors that affect the spatial variation of landscape ecological risk the most;

(4) The interactive influence of land type and CCI, land type, and shortest distance has a greater impact on landscape ecological risk than the synergy of the other two factors.

In recent years, Fuzhou has attached great importance to urban traffic construction, but neglected the negative impact of roads on the ecological environment, resulting in the aggravation of regional ecological risks. Our research shows that with the increase of the road network area, the corridor cutting effect of urban roads will become significant, and the impact of road cutting will gradually penetrate into the landscape patches, which will hinder the energy and information transmission between the source areas, and seriously affect the stability of the urban ecological environment and the habitat and survival of animals. We suggest that in the urban construction planning, we should focus on the site selection of road construction, especially in cities with rich landscape resources, such as Fuzhou. Additionally, blind road laying will have an irreversible impact on the ecology. The forest land, cultivated land, and grassland in Fuzhou are most seriously affected by road cutting, so we should focus on protecting these three types of land to avoid direct crossing of road projects. At the same time, diversified vertical greening and slope greening can also effectively alleviate the cutting damage of roads to the ecological environment.

**Author Contributions:** Conceptualization, Z.Y. and N.Y.; methodology, Z.Y. and L.W.; software, Z.Y. and N.Y.; validation, N.Y., L.W. and C.L.; formal analysis, L.W. and C.L.; investigation, Z.Y. and L.W.; resources, N.Y. and L.W.; data curation, Z.Y.; writing—original draft preparation, Z.Y. and N.Y.; writing—review and editing, Z.Y., N.Y., L.W. and C.L.; visualization, Z.Y. and N.Y.; supervision, N.Y. and L.W.; project administration, N.Y.; funding acquisition, N.Y. All authors have read and agreed to the published version of the manuscript.

**Funding:** This work was supported by the Fujian Province Young and Middle-aged Teacher Education Research Project (JAT210071).

**Institutional Review Board Statement:** Not applicable.

**Informed Consent Statement:** Not applicable.

**Data Availability Statement:** Not applicable.

**Conflicts of Interest:** The authors declare no conflict of interest.

**Appendix A**

**Table A1.** Percentage of cutting area and length of adjoining sides in different road cutting modes.

| | Cutting Mode | Time | Land Use Type | | | | | | |
|---|---|---|---|---|---|---|---|---|---|
| | | | Woodland | Grassland | Cultivated Land | Water Body | Wetland | Building Land | Unused Land |
| Cutting area (%) | Edge cutting | 2000 | 46.34 | 6.42 | 36.99 | 3.18 | 0.76 | 5.93 | 0.39 |
| | | 2010 | 52.09 | 6.47 | 32.12 | 3.06 | 0.92 | 5.02 | 0.32 |
| | | 2020 | 55.58 | 6.38 | 26.35 | 3.32 | 0.80 | 7.34 | 0.24 |
| | Intermediate cutting | 2000 | 21.57 | 6.06 | 49.16 | 5.66 | 1.44 | 14.81 | 1.32 |
| | | 2010 | 27.74 | 6.56 | 43.40 | 4.64 | 1.06 | 15.66 | 0.95 |
| | | 2020 | 33.61 | 5.95 | 35.24 | 5.51 | 0.65 | 18.45 | 0.59 |
| | Complete cutting | 2000 | 10.62 | 3.92 | 31.72 | 12.91 | 2.53 | 34.28 | 4.02 |
| | | 2010 | 13.91 | 4.13 | 31.94 | 10.53 | 1.75 | 33.96 | 3.78 |
| | | 2020 | 15.85 | 6.52 | 26.94 | 13.53 | 1.27 | 33.43 | 2.45 |
| The adjacent side length (%) | Edge cutting | 2000 | 46.35 | 17.09 | 26.46 | 3.23 | 0.84 | 4.47 | 1.56 |
| | | 2010 | 49.04 | 17.22 | 24.92 | 3.09 | 0.93 | 3.51 | 1.29 |
| | | 2020 | 49.53 | 17.62 | 23.52 | 2.79 | 0.81 | 4.65 | 1.08 |
| | Intermediate cutting | 2000 | 22.02 | 19.86 | 32.89 | 6.62 | 1.50 | 11.74 | 5.36 |
| | | 2010 | 25.91 | 21.20 | 29.93 | 6.00 | 1.07 | 11.71 | 4.20 |
| | | 2020 | 30.24 | 20.66 | 28.36 | 4.95 | 0.68 | 12.13 | 2.98 |
| | Complete cutting | 2000 | 8.69 | 10.52 | 17.67 | 17.50 | 3.29 | 28.54 | 13.79 |
| | | 2010 | 11.91 | 11.96 | 19.13 | 14.55 | 2.36 | 25.50 | 14.59 |
| | | 2020 | 16.46 | 19.03 | 18.77 | 13.77 | 1.88 | 20.26 | 9.83 |
| CCI (%) | | 2000 | 10.57 | 39.98 | 36.71 | 4.77 | 1.34 | 5.43 | 1.19 |
| | | 2010 | 12.12 | 37.35 | 38.45 | 4.26 | 1.31 | 5.62 | 0.90 |
| | | 2020 | 17.39 | 33.81 | 38.91 | 3.63 | 1.16 | 4.94 | 0.15 |

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
