# Peer review of "Assessing the Impact of Road Network on Urban Landscape Ecological Risk Based on Corridor Cutting Degree Model in Fuzhou, China"

_sustainability, doi:10.3390/su15021724_

Round 1
Reviewer 1 Report
In this paper, the authors assess the potential implication of the road systems on the landscape ecological risk by taking Fuzhou city as a case. And the study introduced a corridor cutting degree model and analyzed the landscape pattern change from the road system. Finally, the main determinants of landscape eco-logical risk were identified based on a spatial auto-correlation analysis and a geographical detector model. I think this paper is very well structured, organized, and explicit. Then I recommend acceptance after minor revision by modifying a few spells and format.
Author Response
Dear Reviewer,
Thank you very much for your valuable comments. We revised the word order and structure of the paper according to the comments of the reviewers, and improved the shortcomings of the paper through the guidance of professionals
These suggestions have played an important role in my revision of the manuscript.
Thank you for your suggestions.
Best regards,
Authors
Author Response
Dear Reviewer,
We are really grateful for the valuable comments. According to the suggestions of the reviewer, we revised the manuscript. Next, I will reply one by one according to the specific comments of the reviewers. Please see the attachment.
These suggestions have played an important role in my revision of the manuscript.
Thank you for your suggestions.
Best regards,
Authors

Reviewer 3 Report
Dear Authors, thanks for interesting article. The topic of the study seems to be current and interesting.
On the other hand, there are many studies on this topic, so I would like to ask the authors to make an effort to produce moderate revision. The introduction section should made internationally relevant. The Authors focused mainly on studies from Asia/China. Also the conclusions need more than that. It is necessary to show why the applied methodology can be forward-looking in non-Asian road-network ecology. Moreover, the universal results should be highlighted for non-Asian readers.
Scientific assessment of ecological security is essential for protecting regional eco-environment and promoting sustainable development. The manuscript is nicely structured, and the research is solid, with conclusions logically stemming from the results and their interpretations. The topic analysed is in line with some of Sustainability`s themes of interest. This article has a reliable and solid scientific basis and for this reason I recommend to publish them after a moderate revision.
Author Response
Dear Reviewer,
Thank you very much for your valuable comments. According to the suggestions of the reviewers, we first revised the whole paper to make the theme more novel. Secondly, in order to strengthen the universality of the results of the paper, we added an empirical basis for the impact of roads on land space in the introduction, and strengthened the forward-looking basis for the application of our methods in Asia/China. In the discussion section, we also revised the conclusions to strengthen the link between our research results and China's urban land planning.
These suggestions have played an important role in my revision of the manuscript.
Thank you for your suggestions.
Best regards,
Authors
Round 2
Reviewer 2 Report
Dear Authors,
I appreciate your through revision of the paper. I am satisfied with the current version, however please consider comments:
---
# There are still many formatting issues. In some cases you have used two digits after decimal places, but in other more than that. Better to be consistent.
# Line 24: "Corridor cutting degree index" - first letter should be in Capital.
# Line 88-99: Currently, this reads like conclusion. I suggest you to rewrite the phrase as "the necessity of research/study". Line 96: "this study revealed" should not be in the introduction section.
# Conclusion: It is good to point out important findings here but this is too long. Please summarise Lines 609-635 in few sentences.
# Appendix A: "Glassland" >> "Grassland"
---
Lastly, I congratulate on your impressive research work.
Reviewer
Author Response
Dear Reviewer,
Thank you very much for your consideration. According to the comments of reviewers, we make the following modifications:
---
# In terms of format, we uniformly use two decimal places to keep the whole text consistent. The data in Figure 9, Table 4, Table 5 and Table 6 are changed to two decimal places.
# Line 24: "Corridor cutting degree index" - first letter have been modified.
# Line 88-99: We rewrite the phrase as "the necessity of research/study". Revise the words in this paragraph to highlight the innovation and research significance of the article.
# Conclusion: The conclusion part is abridged and summarized with concise sentences.
# Appendix A: We modified the errors in the table.
---
Finally, thank you again for their valuable suggestions.
Authors